# Positive Effects of Saliva on Oral Candidiasis: Basic Research on the Analysis of Salivary Properties

**DOI:** 10.3390/jcm10040812

**Published:** 2021-02-17

**Authors:** Norishige Kawanishi, Noriyuki Hoshi, Takuya Adachi, Narumi Ichigaya, Katsuhiko Kimoto

**Affiliations:** Department of Oral Interdisciplinary Medicine, Prosthodontics & Oral Implantology, Graduate School of Dentistry, Kanagawa Dental University, Yokosuka 238-8580, Japan; kawanishi@kdu.ac.jp (N.K.); adachi@kdu.ac.jp (T.A.); n.hirai@kdu.ac.jp (N.I.); k.kimoto@kdu.ac.jp (K.K.)

**Keywords:** *Candida albicans*, resting salivary flow rate, stimulated salivary flow rate, prosthodontics, denture

## Abstract

The major causes of oral candidiasis include decreased salivary flow rate and the use of ill-fitting dentures. However, the relationships among prosthetic treatment, saliva, and *Candida albicans* have not been elucidated. This study aimed to examine the effects of prosthetic treatment and changes in saliva (mainly the salivary flow rate) on oral candidiasis symptoms. Participants requiring prosthetic treatment underwent testing for *C. albicans*, salivary flow rate, intraoral symptoms, and bite force at the initial visit and four months after treatment to evaluate pretreatment and post-treatment changes. The relationships among *C. albicans*, salivary flow rate, dentures, and intraoral symptoms were analyzed using multiple regression analysis. Denture treatment improved activity against *C. albicans* as well as the salivary flow rate, intraoral symptoms, and masticatory function. Multiple regression analysis revealed that changes in the stimulated salivary flow rate due to prosthetic treatment significantly improved *C. albicans* detection (*p* = 0.011), intraoral symptoms (*p* = 0.037), and bite force (*p* = 0.031). This study showed that prosthetic treatment improved salivary flow and intraoral symptoms and confirmed the influence of stimulated salivary flow rate changes.

## 1. Introduction

Major causes of oral candidiasis include a decreased salivary flow rate, the use of ill-fitting dentures, a poor oral environment, long-term use of antibiotics and immunosuppressants, and chronic diseases such as diabetes mellitus [1,2]. Due to aging societies, increased life expectancy, and the growing prevalence of chronic diseases, the wearing of dentures has increased. Therefore, the number of patients with oral candidiasis is expected to rise. Treatment of oral candidiasis includes improvement in oral hygiene, treatment for ill-fitting dentures, and administration of antifungal agents. As the effect of short-term use of antifungal agents only lasts for a brief period, long-term administration of antifungal agents is necessary in terms of prophylaxis. However, many elderly people have an underlying disease that may require several medications, with a higher risk of side-effects of antifungal agents [3,4].

The reported causes of a reduced salivary flow rate include poor lifestyle, age-related changes, metabolic and endocrine disorders due to systemic disease, and drug therapy [5,6]. A persistently reduced salivary flow rate below the normal range due to reduced oral function, infection, and aspiration pneumonia is known to affect a patient’s general condition [7,8,9]. A reduced salivary flow rate leads to an increased oral *Candida albicans* (*C. albicans)* count [10,11] and the development of oral candidiasis, an opportunistic infection. Saliva contains enzymes, hormones, antibodies, antimicrobial components, and proteins and is affected by diseases and bacteria [12,13]. Previous studies have reported that these salivary components inhibit the growth of *C. albicans* [14]. Reduction in salivary flow and changes in salivary components are major factors affecting oral candidiasis [10,15,16,17].

The use of ill-fitting dentures leads to oral candidiasis and has been associated with denture stomatitis, which affects saliva production and leads to the development of symptoms [4,18,19,20,21,22]. Many studies have reported that prosthetic treatment improved the salivary flow rate [23,24,25,26]. Previously, we investigated the relationship between the salivary flow rate and prosthetic treatment, focusing on oral discomfort such as oral dryness, and found that an improved salivary flow rate led to improved intraoral symptoms. Furthermore, we confirmed that not only prosthetic treatment but also changes in the stimulated salivary flow rate affected oral candidiasis [27,28]. Compared with resting saliva, stimulated saliva has been reported to show greater changes in salivary flow rate and salivary protein concentrations were found to be lower, but concentrations of ions such as sodium and bicarbonate were higher [29]. Stimulated saliva also shows greater changes in salivary metabolites in response to masticatory activity [30].

Previously, we reported that an increase and improvement in salivary flow rate may reduce oral candidiasis [31]. However, not only changes in the salivary flow rate but also salivary composition may affect oral candidiasis. To investigate salivary components, it is necessary to first investigate the relationship between saliva and other factors. This study aimed to identify factors affecting the salivary flow rate and use multiple regression analysis to determine factors that significantly affect oral candidiasis.

## 2. Materials and Methods

### 2.1. Participants

The participants in this study were 62 patients (25 men and 37 women) with dentures who were treated at Kanagawa Dental University Hospital, Kanagawa, Japan and who had been assessed to be in need of new dentures by a prosthetist at the initial visit to improve occlusal abnormalities. All examinations were performed at the time of the first visit and after completion of the denture treatment. Denture treatment ended approximately four months after new dentures had been fitted and adjusted, and once the dentures were stabilized. Denture treatment comprised fabrication of new dentures by a prosthodontist for all participants to improve occlusion. Those suffering from dementia or other psychiatric disorders that rendered communication difficult at the time of participant selection were excluded. Informed consent was obtained from all participants, and the study protocol was reviewed and approved by the Ethics Committees of Kanagawa Dental University (approval 2016.6.1 number 588).

### 2.2. Detection of C. albicans

*C. albicans* was detected using Stomastat (Sankin Industry Co. Ltd., Osaka, Japan) at the initial visit. Samples were collected by scratching the buccal mucosa with a sterile cotton swab and were incubated at 37 °C for 24 h. Based on their test results, participants were divided into two groups, namely, a control group, comprising those without *Candida* infection (25 participants; 10 men and 15 women), and a *Candida* group, comprising those with *Candida* infection (37 participants; 13 men and 24 women). During this study, the *Candida* group did not receive drug therapy.

### 2.3. Measurement Items

All measurements were performed twice: at the initial visit and four months after successful adjustment of the new dentures.

#### 2.3.1. Salivary Flow Rate

The resting salivary flow rate was evaluated using the spitting method. During the collection period (15 min), participants were requested not to swallow. The baseline level for the reduced resting salivary flow rate was set at 1.5 mL/15 min [32,33].

The stimulated salivary flow rate was evaluated using the chewing gum method. During the saliva collection period (10 min), participants were requested not to swallow while chewing gum (Free Zone; Lotte Co., Ltd., Tokyo, Japan). The baseline for the reduced stimulated salivary flow rate was set at 10 mL/10 min [6,34].

The salivary flow rate was measured by the same surgeon. To minimize diurnal variation in the resting salivary flow rate, participants were requested not to eat or drink (except water) 1 h prior to measurement of the salivary flow rate at approximately 10:00 a.m. [35].

#### 2.3.2. Intraoral Symptoms

According to previous studies [28], the presence or absence of the following symptoms were evaluated: the presence of pseudomembrane, subjective tingling (pain and burning sensation), angular cheilitis, angular stomatitis, redness of the buccal mucosa, denture plaques, tongue coating, and taste disorders. The participants were asked to indicate the presence or absence of the above symptoms on a scale, with a total score ranging from 0 to 7 points (presence = 1; absence = 0). The relationship between the obtained scores and *C. albicans* was analyzed [28].

#### 2.3.3. Bite Force Test

Bite force was measured using a color-changing chewing gum (Xylitol Masticatory Performance Evaluating Gum; Lotte, Tokyo, Japan). Bite force was measured through evaluating the change in the color of the chewing gum after chewing [36,37,38,39]. According to the manufacturer’s instructions for denture users, the mastication time was set at 3 min. Subsequently, the color of the chewing gum was classified and scored using a color chart.

Bite force was classified according to the following scores: 1 point, green or yellow (low bite force); 2 points, pink (average bite force); 3 points, red (good bite force).

### 2.4. Statistical Analysis

Chi-square test, Student’s t-test, and Mann–Whitney U test were performed to evaluate the effect of prosthetic treatment on salivary flow rate, intraoral symptoms, and bite force.

Multiple regression analysis was used to evaluate the following hypotheses: (1) denture treatment affects oral candidiasis, intraoral symptoms, bite force, and salivary flow rate and (2) changes in the resting and stimulated salivary flow rates affect oral candidiasis, intraoral symptoms, and bite force. Sex characteristics were used as confounding factors in the multiple regression analysis. The dependent variable was the changes in scores calculated by subtracting the pretreatment scores from the post-treatment scores. A *p*-value < 0.05 was considered statistically significant.

All analyses were performed using IBM^®^ SPSS^®^ Statistics version 21 (IBM Corp., Armonk, NY, USA).

## 3. Results

### 3.1. Participants

The characteristics of the participants are presented in Table 1. The mean age was significantly higher in the *Candida* group than in the control group (*p* < 0.05, unpaired *t*-test). Although not significant, the number of female participants was higher than that of male participants. The results showed that all participants used dentures, with partial dentures being the most common.

Multiple regression analysis was performed using sex, type of dentures, denture treatment, bite force, and changes in salivary flow rate as dependent variables, and changes in scores for the three variables (*C. albicans*, intraoral symptoms, and bite force) as independent variables. Multiple regression analysis was also performed using participants’ sex, prosthetic conditions, and prosthetic treatment as dependent variables, and pretreatment and post-treatment changes in scores for the two variables (resting and stimulated salivary flow rates) as independent variables.

### 3.2. Changes in C. albicans

All participants in the *Candida* group tested positive for the *Candida* test at the initial visit. All participants in the *Candida* group had a significantly lower *C. albicans* count (<100 CFU (Colony Forming Unit)/mL) at four months after treatment than at the initial visit. Histopathological examinations were also performed at both visits. None of the participants in this study received drug therapy for oral candidiasis.

### 3.3. Salivary Flow Rate

The mean resting salivary flow rate at the initial visit was 2.12 ± 0.14 mL/15 min for the control group and 1.77 ± 0.11 mL/15 min for the *Candida* group, and both values were greater than the baseline value (1.5 mL/15 min). In addition, there was a significant difference in the resting salivary flow rates between the two groups (*p* < 0.05, Mann–Whitney U test). Furthermore, no significant difference was observed in either the resting salivary flow rate after treatment between the control (2.44 ± 0.15 mL/15 min) and *Candida* groups (1.97 ± 0.21 mL/15 min; *p* = nonsignificant) or in temporal changes in the resting salivary flow rates in each group (*p* = nonsignificant) (Figure 1).

In the control group, the stimulated salivary flow rate at the initial visit (17.18 ± 2.51 mL/10 min) was greater than the baseline value (10 mL/10 min). Meanwhile, in the *Candida* group, the stimulated salivary flow rate at the initial visit (8.92 ± 2.11 mL/10 mL) was lower than the baseline value. Thus, there was a significant difference in the stimulated salivary flow rate between the two groups (*p* < 0.05, Mann–Whitney U test). In the *Candida* group, the stimulated salivary flow rate after treatment (12.15 ± 2.81 mL/10 mL) was greater than the baseline value, showing a significant improvement compared with the initial visit (*p* < 0.05, paired *t*-test). There was no significant difference in the salivary flow rate after treatment between the two groups (Figure 2).

### 3.4. Intraoral Symptoms

The oral symptom score at the initial visit was significantly higher in the *Candida* group than in the control group (5.99 ± 0.81 vs. 3.11 ± 0.55; *p* < 0.05, Mann–Whitney U test), indicating the presence of oral candidiasis symptoms in the *Candida* group.

In both groups, the oral symptom score after treatment was significantly lower than that at the initial visit (control group, 0.56 ± 0.18; *Candida* group, 1.11 ± 0.25; *p* < 0.05 for both groups, paired *t*-test). The symptom score was significantly higher in the *Candida* group than in the control group, but both groups showed a tendency toward improvement in symptom score (*p* < 0.05, Mann–Whitney U test) (Figure 3).

### 3.5. Bite Force

In the control group, the mean score for bite force at the initial visit was 2.69 ± 0.11, showing a good bite force. Meanwhile, in the *Candida* group, the mean score for bite force at the initial visit was 1.01 ± 0.41, showing a reduced bite force (*p* < 0.05, Mann–Whitney U test).

In both groups, the mean score for bite force after treatment was ≥2.5, with a significant improvement in bite force in the *Candida* group (*p* < 0.05, paired t-test) (Figure 4).

### 3.6. Multiple Regression Analysis of the Salivary Flow Rate, Intraoral Symptoms, and Bite Force

Table 2 shows the results of the multiple regression analysis with sex, type of dentures, denture treatment, bite force, and changes in salivary flow rate as independent variables and changes in scores for the three variables (*C. albicans*, intraoral symptoms, and bite force) as dependent variables. Additionally, Table 2 shows the results of multiple regression analysis with sex, prosthetic condition, and prosthetic treatment as independent variables and changes in scores for two variables (resting and stimulated salivary flow rate) before and after treatment as dependent variables.

The analysis identified sex (*p* = 0.029), type of dentures (*p* = 0.012), prosthetic treatment (*p* = 0.01), changes in bite force (*p* = 0.025), and changes in the stimulated salivary flow rate (*p* = 0.011) as factors strongly affecting the detection of *C. albicans*. The analysis also identified type of dentures (*p* = 0.072), changes in bite force (*p* = 0.042), and changes in the stimulated salivary flow rate (*p* = 0.037) as factors strongly affecting improved intraoral symptoms. In addition, prosthetic treatment (*p* = 0.048) and changes in the stimulated salivary flow rate (*p* = 0.031) were identified as factors strongly affecting improved bite force.

The resting salivary flow rate and the changes in the stimulated salivary flow rate were only associated with prosthetic treatment (resting salivary flow rate, *p* = 0.041; stimulated salivary flow rate, *p* = 0.008).

## 4. Discussion

This study examined the effects of prosthetic treatment on oral candidiasis. All the factors considered (i.e., *C. albicans* count, changes in salivary flow rate, intraoral symptoms, and bite force) showed significant improvement in pre- and post-prosthetic treatment comparisons. Therefore, the relationship between oral candidiasis and prosthetic treatment was further analyzed using a multiple regression analysis. The results showed that prosthetic treatment improved the salivary flow rate and that changes in the stimulated salivary flow rate particularly affected the symptoms of oral candidiasis.

In patients with dentures, the growth of *C. albicans* leads to denture stomatitis [20,22,40]. Patients with ill-fitting or inadequately cleaned dentures have been reported to have a higher risk of denture stomatitis due to denture plaques [41]. Complications such as inflammation and redness of the buccal mucosa or ulcer formation lead to dysfunction due to dentures, resulting in decreased bite force and salivary flow rate [42]. This study demonstrated that prosthetic treatment improved the salivary flow rate and reduced the *C. albicans* count, thus improving intraoral symptoms. Increased and improved salivary flow rates due to the use of dentures have been reported [25,43]. Previously, we reported that prosthetic treatment may improve the salivary flow rate and hence intraoral symptoms [28]. The results of this study are consistent with those of previous studies.

The multiple regression analysis suggested that changes in the stimulated salivary flow rate strongly affected prosthetic treatment and the *C. albicans* count and improved intraoral symptoms and bite force. Therefore, the development of new dentures may increase the stimulated salivary flow rate, thus reducing the *C. albicans* count and ameliorating symptoms of oral candidiasis. The results of this study are consistent with those of previous studies that have shown a significant relationship between the stimulated salivary flow rate and the *C. albicans* count [10,44,45].

This study had some limitations. Although we conducted the *C. albicans* test using Stomastat, we did not examine other bacterial species. It has been reported that not only single bacterial species, such as *C. albicans*, but also mixed bacterial species, such as *Candida glabrata*, *Candida parapsilosis*, and *Candida tropicalis*, can be found among the *Candida* species on dentures [46]. Further studies concerning the effects of each bacterial species or mixed bacterial species may be necessary. Furthermore, diurnal variation in the number of detected *C. albicans* and prosthetic conditions in patients with dentures have been reported [47]. The effect of difference in collection sites (e.g., the buccal mucosa or the inner surface of dentures) of *C. albicans* and the time of collection need to be reconsidered. The study results showed that the stimulated salivary flow rate affected oral symptoms and the *C. albicans* count. However, it is speculated that the improvement in oral function due to denture treatment contributed to masticatory efficiency and promoted an increase in the stimulated salivary flow rate. In this study, only the change in saliva volume was investigated. Further investigation of saliva composition may be necessary to identify other influencing factors.

Saliva has antibiotic properties and several other functions including providing mucous membrane protection, buffer action, tooth remineralization, digestive effects, and a cleaning effect. Salivary flow is also an important indicator of salivary changes, although the role of salivary components is significant. In particular, factors such as lysozyme, peroxidase, lactoferrin, histatin, and secretory immunoglobulin A have antibiotic-like properties. Previous studies have reported the effect of salivary cytokines as biomarkers on denture stomatitis [48]. In addition, saliva contains various substances such as metabolites, proteins, mRNA, DNA, enzymes, hormones, antibodies, antimicrobial components, and growth factors. Changes in these substances can be affected by intraoral and systemic conditions. Recently, metabolomic analysis, a comprehensive in vivo analysis of metabolites, has attracted attention [49]. In particular, studies using metabolomic analysis of saliva [29,49,50,51,52,53] have shown significant effects of biomarker diagnosis in the treatment of oral and systemic cancers [54]. Therefore, studies of salivary flow rate and changes in salivary components may clarify the effects of saliva on oral candidiasis. Saliva has been reported to contain various proteins that affect *C. albicans* [14]. Moreover, studies of the effect of saliva on *C. albicans* have identified a significant effect on salivary cytokines and proteins [14]. In our previous studies using metabolomic analysis of saliva, we reported significant differences in substances in metabolites of resting and stimulated saliva [30]. The above findings and a significant increase, or improvement, in the stimulated salivary flow rate in the *Candida* group in this study draw attention not only to the cleaning effect of saliva but also to the effect of salivary components on oral candidiasis. Therefore, the salivary flow rate and salivary components may be important factors affecting symptoms of intraoral conditions such as oral candidiasis.

While the study demonstrated that prosthetic treatment improved salivary flow rate and that stimulated saliva has an effect on the symptoms of oral candidiasis, this study is not without limitation. The changes in the composition of saliva were not investigated in this study. Analysis of salivary components may help to identify factors that influence *C. albicans*. In investigating the effects of salivary components, changes in saliva volume, especially the stimulated salivary flow rate with large quantitative changes, should be monitored in future studies.

## 5. Conclusions

In this study, the effect of prosthetic treatment on oral candidiasis was examined using multiple regression analysis of the salivary flow rate, the *C. albicans* count, and intraoral symptoms. The results confirmed the strong effect of stimulated saliva on the symptoms of oral candidiasis.

## Figures and Tables

**Figure 1 jcm-10-00812-f001:**
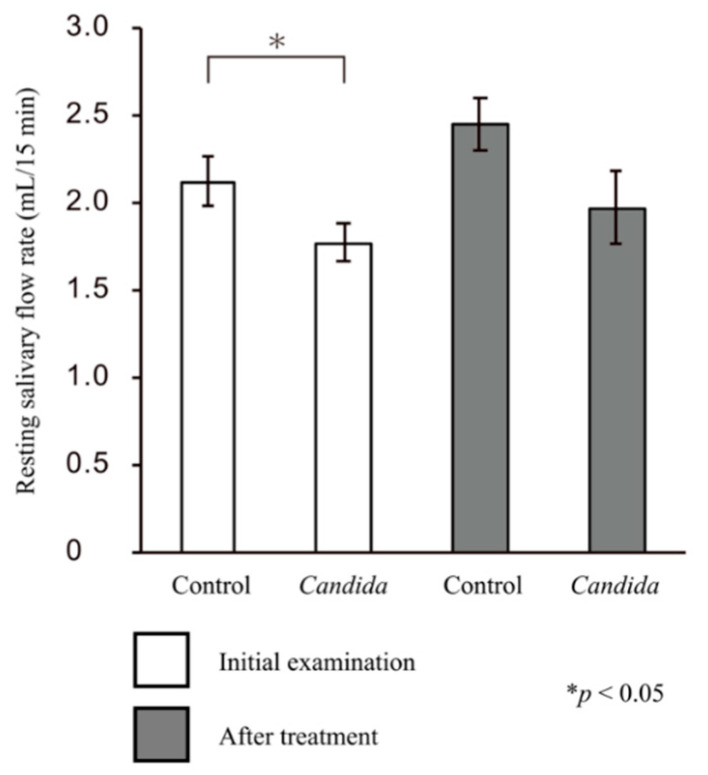
Changes in the resting salivary flow rate. There was a significant difference in the changes in the resting salivary flow rate at the initial visit between the control and *Candida* groups (*p* < 0.05). Although not significant, the changes in the resting salivary flow rate after prosthetic treatment were less in the *Candida* group than in the control group.

**Figure 2 jcm-10-00812-f002:**
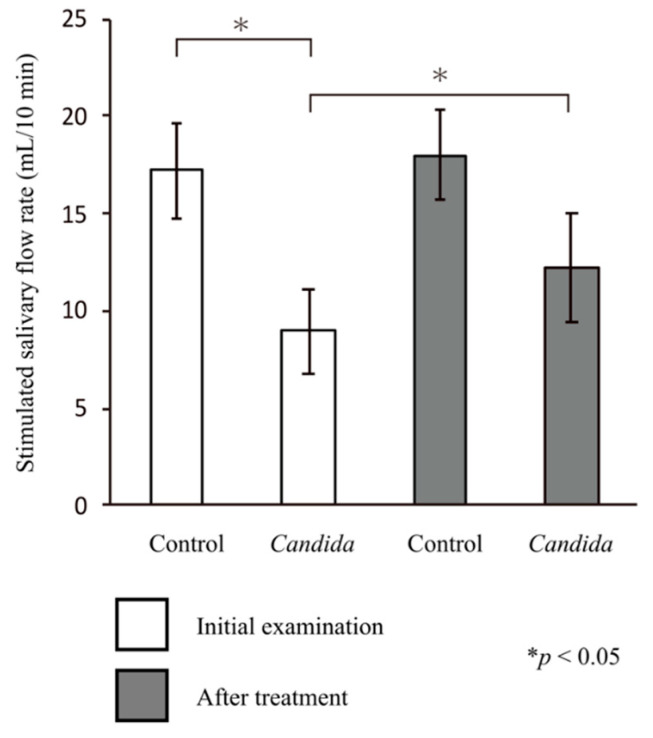
Changes in the stimulated salivary flow rate. There was a significant difference in the changes to the stimulated salivary flow rate at the initial visit between the control and *Candida* groups (*p* < 0.05). In the *Candida* group, the stimulated salivary flow rate after prosthetic treatment increased significantly (*p* < 0.05).

**Figure 3 jcm-10-00812-f003:**
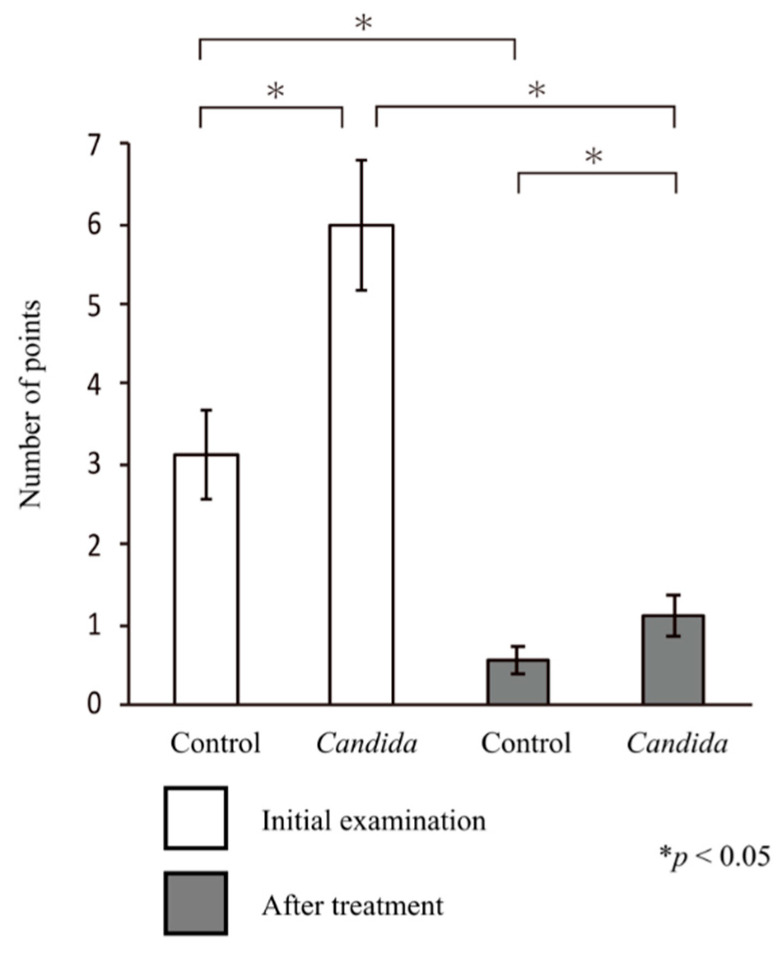
Changes in intraoral symptoms. The analysis of the scores for intraoral symptoms showed significant differences in the changes in intraoral symptoms between the control and *Candida* groups, both at the initial visit and after prosthetic treatment (*p* < 0.05). The results showed significantly improved intraoral symptoms after treatment in both groups (*p* < 0.05).

**Figure 4 jcm-10-00812-f004:**
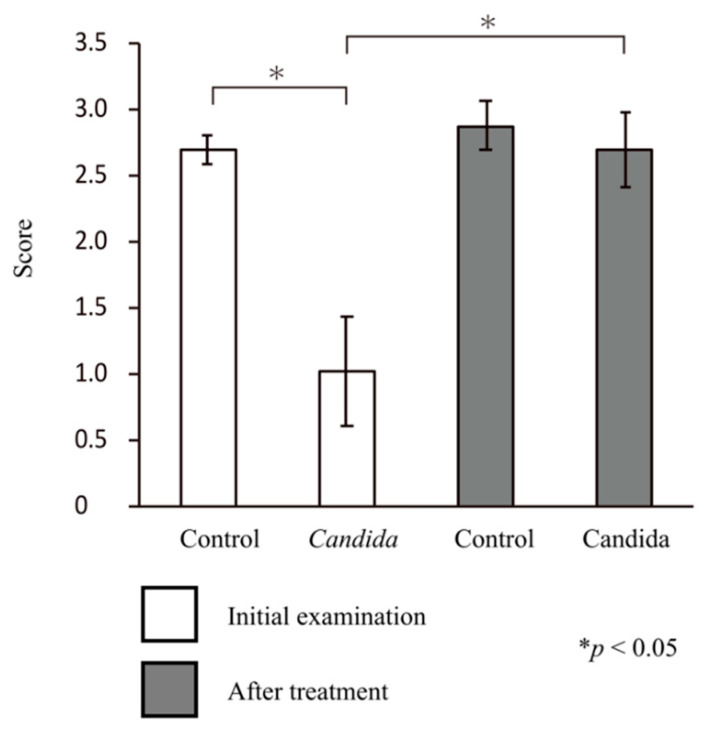
Bite force test. At the initial visit, bite force was significantly lower in the *Candida* group than in the control group (*p* < 0.05). In the *Candida* group, bite force showed a significant improvement in pre- and post-prosthetic treatment comparisons (*p* < 0.05).

**Table 1 jcm-10-00812-t001:** Participant characteristics.

	Control Group (*n* = 25)	*Candida* Group (*n* = 37)
Age (years)	71.1 ± 2.4	74.5 ± 2.8 *
Sex		
Male	10 (40)	13 (35)
Female	15 (60)	24 (65)
Prosthetic		
Partial denture	22	33
Full denture	13	29
Prosthetic treatment (%)	100	100

Values are presented as mean ± standard deviation or *n* (%), * *p* < 0.05.

**Table 2 jcm-10-00812-t002:** Multiple regression analysis.

Dependent Variable	Independent Variable	Non-Standardization Factor	*p*-Value
*Candida*	Sex (male/female)	1.511	0.029 *
	Type of denture (PD/FD)	−0.428	0.012 *
	Existing denture treatment (new/old)	−2.481	0.01 *
	Masticatory ability	1.922	0.025 *
	Unstimulated salivary	0.399	0.701
	Stimulated salivary	−0.459	0.011 *
Oral symptoms	Sex (male/female)	0.819	0.059
	Type of denture (PD/FD)	−0.924	0.072 *
	Existing denture treatment (new/old)	1.192	0.104
	Masticatory ability	0.725	0.042 *
	Unstimulated salivary	1.291	0.079
	Stimulated salivary	−0.255	0.037 *
Masticatory ability	Sex (male/female)	1.102	0.318
	Type of denture (PD/FD)	0.982	0.092
	Existing denture treatment (new/old)	1.587	0.048 *
	Unstimulated salivary	0.911	0.441
	Stimulated salivary	1.745	0.031 *
Unstimulated salivary flow rate	Sex (male/female)	0.009	0.329
	Type of denture (PD/FD)	0.155	0.11
	Existing denture treatment (new/old)	0.75	0.041 *
Stimulated salivary flow rate	Sex (male/female)	−0.075	0.212
	Type of denture (PD/FD)	1.102	0.059
	Existing denture treatment (new/old)	1.029	0.008 *

* *p* < 0.05, PD, partial denture; FD, full denture.

## Data Availability

The datasets generated and analyzed during the current study are available from the corresponding author on reasonable request.

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
