# Peer review of "Positive Effects of Saliva on Oral Candidiasis: Basic Research on the Analysis of Salivary Properties"

_jcm, 2021, doi:10.3390/jcm10040812_

Round 1

Reviewer 1 Report

This is a paper with interesting conclusions, but several aspects need to be clarified.

First of all, the title of this paper mentions that this is a metabolomic study, but this is not the case. Please remove that it is a metabolomic study.

The wording and use of English needs to be improved. Some sentences sound good, but at close reading don’t say anything. I recommend the authors to carefully reread the manuscript.

Overall, the authors strongly suggest that changes in salivary flow is one of the key factors in improving intraoral symptoms. It is difficult to conclude whether the increase in salivary flow rate actually contributes to improvement of intraoral symptoms. It may also be the other way around, the stimulated salivary flow rate may improve, because patients are better able to chew and stimulate their salivary flow in that way. Unless this is proven in some way, authors should remove suggestive sentences like the last sentence in the abstract ‘This study showed that prosthetic treatment improves the salivary flow rate and intraoral symptoms, mainly through changes in the stimulated salivary flow rate.

The abstract clearly defines the research question and summarizes the results. Denture treatment improves activity against Candida albicans, salivary flow rate, clinical outcome and masticatory function.   

Introduction line 39. the causes of reduced salivary flow rate may be reduced salivary flow rate. This is an awkward sentence.

Line 47-48  studies on salivary components identified the reductions and changes in salivary flow rate. Flow rate is not a component.

line 58-59 Stimulated saliva contains a high concentration of salivary components. This suggests that the concentration of components like proteins is higher in stimulated saliva compared to unstimulated saliva. This is not true. Only some ions like sodium and bicarbonate are higher, but the concentration of most proteins is lower in stimulated saliva compared to unstimulated saliva.

Materials and methods.

Detection of Candida albicans. It is unclear how samples for Candida detection are taken’, please describe.

Results.

The results are clearly presented, but carefully look at the description of dependent and independent variables . In line 201 the authors write ‘Table 2 shows the results of the multiple regression analysis with sex, type of dentures ... as dependent variables’ This must be independent variables. Sex or gender never can be a dependent variable. This should have as a consequence that gender changes under the influence of treatment. In Table 2 please speak of ‘ ‘salivary flow rate’ instead of only ‘salivary’.

Line 217-220 repeat line 202-205 with again a wrong definition of dependent and independent variables. Please remove. 

line 247 ‘have supported a significant relationship’ must be ‘have shown a significant relationship’

Line 259. Although the changes in the salivary flow rate are an important predictor of salivary activity..What is meant with this sentence?

Author Response

The manuscript has been rechecked and the necessary changes have been made in accordance with the reviewers’ suggestions. The responses to all comments have been prepared and attached herewith. 

First of all, the title of this paper mentions that this is a metabolomic study, but this is not the case. Please remove that it is a metabolomic study.

Response: Thank you for your suggestion. The title has been revised (Line 2).

The wording and use of English needs to be improved. Some sentences sound good, but at close reading don’t say anything. I recommend the authors to carefully reread the manuscript.

Response: Thank you for your comments. Our manuscript has undergone another thorough editing and proofreading round and has been amended accordingly.

Overall, the authors strongly suggest that changes in salivary flow is one of the key factors in improving intraoral symptoms. It is difficult to conclude whether the increase in salivary flow rate actually contributes to improvement of intraoral symptoms. It may also be the other way around, the stimulated salivary flow rate may improve, because patients are better able to chew and stimulate their salivary flow in that way. Unless this is proven in some way, authors should remove suggestive sentences like the last sentence in the abstract ‘This study showed that prosthetic treatment improves the salivary flow rate and intraoral symptoms, mainly through changes in the stimulated salivary flow rate.

Response : Thank you for your comments. Corrections have been made to the last sentence of the abstract accordingly.

The abstract clearly defines the research question and summarizes the results. Denture treatment improves activity against Candida albicans, salivary flow rate, clinical outcome and masticatory function.   

Introduction line 39. the causes of reduced salivary flow rate may be reduced salivary flow rate. This is an awkward sentence.

Response: Thank you for pointing this out. We have revised the sentence accordingly (Lines 38-40).

Line 47-48  studies on salivary components identified the reductions and changes in salivary flow rate. Flow rate is not a component.Response: The reference has been checked, and the text has been corrected. (Line 57-60)

line 58-59 Stimulated saliva contains a high concentration of salivary components. This suggests that the concentration of components like proteins is higher in stimulated saliva compared to unstimulated saliva. This is not true. Only some ions like sodium and bicarbonate are higher, but the concentration of most proteins is lower in stimulated saliva compared to unstimulated saliva.

Response: Thank you for your valuable comments. The sentence has been amended accordingly (Lines 55-58)

Materials and methods.

Detection of Candida albicans. It is unclear how samples for Candida detection are taken’, please describe.

Response: Thank you for this comment. We have added further details concerning the sample collection process (Line 82-83)

Results.

The results are clearly presented, but carefully look at the description of dependent and independent variables . In line 201 the authors write ‘Table 2 shows the results of the multiple regression analysis with sex, type of dentures ... as dependent variables’ This must be independent variables. Sex or gender never can be a dependent variable. This should have as a consequence that gender changes under the influence of treatment. In Table 2 please speak of ‘ ‘salivary flow rate’ instead of only ‘salivary’.

Response: Thank you for your important comments. We have amended the text in keeping with your comments and have modified Table 2 accordingly.

Line 217-220 repeat line 202-205 with again a wrong definition of dependent and independent variables. Please remove. 

Response: We have modified the text in keeping with your comment (Line 228-234)

line 247 ‘have supported a significant relationship’ must be ‘have shown a significant relationship’

Response: Thank you. The wording has now been amended (Line 262)

Line 259. Although the changes in the salivary flow rate are an important predictor of salivary activity..What is meant with this sentence?

Response: Kindly refer to our revised text on Lines 281-282.

Reviewer 2 Report

This research is under the scope of this journal; the topic is relevant for readers, and this research deals with potentially significant knowledge to the field. The introduction is direct to the objective that you want to study, and this research is well done. The results have been well documented. These results induce this conclusion.

  • However, there are some concerns about the present manuscript: 

Introduction

  • What was your hypothesis null hypothesis?

Materials and Methods

  • Please include a statement in the Material and Methods section that the study has been approved by the institutional ethics committee and provide the number of the process.
  • Made a flowchart, to explain to reads the sequence of the study, and with the inclusion, exclusion criteria.
  • How was the sample calculated? Did the authors perform a power analysis to evaluate if this sample size was appropriate?

Discussion 

  • Please, clarified other limitations of this study?
  • And, clarified the future perspectives.

References

  • References are current, relevant and well inserted.
  • But references are not standardized. The titles of references have a different format, 
    the title of the article is written in capital letters at the beginning of words, others only in lower case. Also, the standardized format of presentation in the journal's name. Because names have written in a different format, one is not abbreviated, others are not.

Author Response

The manuscript has been rechecked and the necessary changes have been made in accordance with the reviewers’ suggestions. The responses to all comments have been prepared and attached herewith. 

Introduction

  • What was your hypothesis null hypothesis?

Response: Corrections have been made to Lines 61-63 accordingly.

Materials and Methods

  • Please include a statement in the Material and Methods section that the study has been approved by the institutional ethics committee and provide the number of the process.

Response: Thank you for drawing this to our attention. We have amended the text accordingly on Lines 78-79.

  • Made a flowchart, to explain to reads the sequence of the study, and with the inclusion, exclusion criteria.

Response: Thank you for your suggestion. Please refer to our modified texts (Lines 70-79).

  •  
  • How was the sample calculated? Did the authors perform a power analysis to evaluate if this sample size was appropriate?Response : The mean difference between the two groups was 2 points. We calculated the sample size based on the standard deviation of the improvement obtained in previous studies, assuming the difference between the mean values of the two groups to be 2 points, with 24 participants in each group, for a total of 48 participants.

Discussion 

  • Please, clarified other limitations of this study?

Response: Thank you for this important comment. Limitations to our study have been added (Lines 303-307)

  • And, clarified the future perspectives.

Response: We have added further details concerning this on Lines 307-309

References

  • References are current, relevant and well inserted.
  • But references are not standardized. The titles of references have a different format, 
    the title of the article is written in capital letters at the beginning of words, others only in lower case. Also, the standardized format of presentation in the journal's name. Because names have written in a different format, one is not abbreviated, others are not.

Response: Thank you for highlighting this. The reference list has been amended accordingly.

Round 2

Reviewer 2 Report

This research is under the scope of this journal; the topic is interesting for readers and this research deals with potentially significant knowledge to the field and an open new way for future studies.

The authors improved the quality of the manuscript after the reviewer's indications.